# Modulation of Plasma Lipidomic Profiles in Metastatic Castration-Resistant Prostate Cancer by Simvastatin

**DOI:** 10.3390/cancers14194792

**Published:** 2022-09-30

**Authors:** Blossom Mak, Hui-Ming Lin, Thy Duong, Kate L. Mahon, Anthony M. Joshua, Martin R. Stockler, Howard Gurney, Francis Parnis, Alison Zhang, Tahlia Scheinberg, Gary Wittert, Lisa M. Butler, David Sullivan, Andrew J. Hoy, Peter J. Meikle, Lisa G. Horvath

**Affiliations:** 1Medical Oncology, Chris O’Brien Lifehouse, Camperdown, NSW 2050, Australia; 2Garvan Institute of Medical Research, Darlinghurst, NSW 2010, Australia; 3Faculty of Medicine and Health, University of Sydney, Camperdown, NSW 2050, Australia; 4St Vincent’s Clinical School, UNSW Sydney, Darlinghurst, NSW 2010, Australia; 5Metabolomics Laboratory, Baker Heart and Diabetes Institute, Melbourne, VIC 3004, Australia; 6Royal Prince Alfred Hospital, Camperdown, NSW 2050, Australia; 7Kinghorn Cancer Centre, St Vincent’s Hospital, Darlinghurst, NSW 2010, Australia; 8Concord Cancer Centre, Concord Repatriation General Hospital, Concord, NSW 2139, Australia; 9Faculty of Medicine and Health Sciences, Macquarie University, Macquarie Park, NSW 2109, Australia; 10Adelaide Cancer Centre, Kurralta Park, SA 5037, Australia; 11South Australian Immunogenomics Cancer Institute and Freemasons Centre for Male Health and Wellbeing, University of Adelaide, Adelaide, SA 5005, Australia; 12South Australian Health and Medical Research Institute, Adelaide, SA 5000, Australia; 13NSW Health Pathology, Department of Chemical Pathology, Royal Prince Alfred Hospital, Camperdown, NSW 2050, Australia; 14School of Medical Sciences, Charles Perkins Centre, University of Sydney, Sydney, NSW 2050, Australia; 15Department of Cardiovascular Research Translation and Implementation, La Trobe University, Bundoora, VIC 3086, Australia

**Keywords:** ceramides, metabolic therapy, lipidomic, prostate cancer, sphingolipids, statins

## Abstract

**Simple Summary:**

Men with metastatic castration-resistant prostate cancer (mCRPC) have shorter overall survival and resist therapy faster if their blood have a poor lipid profile. This poor lipid profile includes high levels of sphingolipids, thus reducing these sphingolipids may slow prostate cancer growth. The aim of our study is to determine if simvastatin can change a poor lipid profile (high sphingolipids) into a better profile (low sphingolipids) in mCRPC. Twenty-seven men with mCRPC were given simvastatin together with their standard treatment for 12 weeks. We found that 11 men had the poor lipid profile at the start of the study. After 12 weeks of treatment, 40% of these 11 men no longer had the poor lipid profile and their sphingolipids were reduced, regardless of changes in their blood cholesterol, LDL or triglycerides. In conclusion, simvastatin can modify the poor lipid profile in some men with mCRPC.

**Abstract:**

Elevated circulating sphingolipids are associated with shorter overall survival and therapeutic resistance in metastatic castration-resistant prostate cancer (mCRPC), suggesting that perturbations in sphingolipid metabolism promotes prostate cancer growth. This study assessed whether addition of simvastatin to standard treatment for mCRPC can modify a poor prognostic circulating lipidomic profile represented by a validated 3-lipid signature (3LS). Men with mCRPC (*n* = 27) who were not on a lipid-lowering agent, were given simvastatin for 12 weeks (40 mg orally, once daily) with commencement of standard treatment. Lipidomic profiling was performed on their plasma sampled at baseline and after 12 weeks of treatment. Only 11 men had the poor prognostic 3LS at baseline, of whom five (45%) did not retain the 3LS after simvastatin treatment (expected conversion rate with standard treatment = 19%). At baseline, the plasma profiles of men with the 3LS displayed higher levels (*p* < 0.05) of sphingolipids (ceramides, hexosylceramides and sphingomyelins) than those of men without the 3LS. These plasma sphingolipids were reduced after statin treatment in men who lost the 3LS (mean decrease: 23–52%, *p* < 0.05), but not in men with persistent 3LS, and were independent of changes to plasma cholesterol, LDL-C or triacylglycerol. In conclusion, simvastatin in addition to standard treatment can modify the poor prognostic circulating lipidomic profile in mCRPC into a more favourable profile at twice the expected conversion rate.

## 1. Introduction

The latest global cancer statistics by the World Health Organisation ranked prostate cancer as the second most frequent cancer and fifth leading cause of cancer death among men worldwide [1]. The survival of men with metastatic castration-resistant prostate cancer (mCRPC) have significantly improved with the discovery of various therapies such as taxane chemotherapy, androgen receptor signalling inhibitors (ARSI), poly-ADP ribose polymerase inhibitors and targeted radioisotopes. However, long term control of mCRPC requires “a combined approach targeting multiple hallmarks of cancer, encompassing the cancer genome, the immune system and metabolic factors including lipid metabolism, all of which contribute to cancer progression and treatment resistance” [2,3]. 

Previously we discovered that elevated levels of circulating sphingolipids were associated with poorer clinical outcomes across the natural history of prostate cancer, e.g., higher rates of metastatic relapse in localised disease, earlier androgen deprivation failure in metastatic hormone-sensitive disease, and shorter overall survival (OS) in mCRPC [4,5,6]. A poor prognostic circulating 3-lipid signature (3LS) was derived and validated in internal and external validation cohorts, and was independently associated with shorter radiographic progression-free survival and OS in patients with mCRPC commencing standard-of-care taxanes or ARSI [3,4,5]. The 3LS consists of two sphingolipids [ceramide(d18:1/24:1), sphingomyelin(d18:2/16:0)] and a glycerophospholipid [phosphatidylcholine(16:0/16:0)] [4]. Higher plasma levels of all three species were previously associated with shorter OS in mCRPC [3,4,5]. However, it remains to be determined if circulating sphingolipids contribute to prostate cancer progression, and if modulating the circulating levels of sphingolipids or the 3LS will improve clinical outcomes. 

Statins, the cholesterol-lowering medication, are able to significantly reduce the plasma levels of sphingolipids, including ceramides and sphingomyelin in patients with cardiovascular disease [7,8,9,10]. Large epidemiological studies have shown that statin use in prostate cancer was associated with better clinical outcomes such as longer time to recurrence, improved OS and reduced prostate cancer-specific mortality [11,12,13,14]. For example, OS for mCRPC was prolonged in those treated with statins in the COU-AA-302 and COU-AA-301 studies which demonstrated a survival benefit for the use of abiraterone before or after docetaxel chemotherapy [15]. Overall, these observations suggest that statin therapy may improve the clinical outcomes of men with mCRPC by modulating the plasma levels of lipids. We hypothesise that statin therapy may be even more effective in improving clinical outcomes if selectively used in mCRPC patients with elevated circulating ceramides or a poor prognostic lipid profile represented by the 3LS. The first step is to determine whether statin treatment can alter the poor prognostic circulating lipid profile of men with mCRPC.

The aim of this study was to investigate whether the addition of simvastatin to standard treatment for mCRPC modulates a poor prognostic circulating lipidomic profile represented by the validated 3-lipid signature.

## 2. Materials and Methods

### 2.1. Study Design and Population

This investigator-initiated, multi-centre, single arm, pilot study enrolled patients from 5 centres in Sydney and Adelaide, Australia. Sample size was calculated prospectively—assuming 25% of patients have the poor prognosis 3LS at baseline [4], a total sample size of 60 patients provides over 90% power (1-sided Type 1 error = 10%) to detect the conversion of the poor prognostic signature in 50% of patients.

Patients with mCRPC were eligible for this study if they were commencing taxane chemotherapy (docetaxel or cabazitaxel) or ARSI (enzalutamide or abiraterone) for disease progression, and not on a lipid-lowering agent. Detailed inclusion and exclusion criteria are listed in Appendix A. 

Participants were treated with simvastatin 40 mg orally once daily for 12 weeks, commencing on day 1 of treatment for mCRPC (Figure 1A). Potential side effects were assessed during 3-weekly visits while on simvastatin and at the follow-up safety visit (3–6 weeks following completion of simvastatin treatment). All participants provided written informed consent. The protocol and all amendments were approved by the Human Research Ethics Committee of the Sydney Local Health District (Ethics Approval No. CH62/6/2017-063) and abided by the Declaration of Helsinki principles. The trial is registered in the Australian New Zealand Clinical Trials Registry (Registration No. ACTRN 12617000965303; Registration date: 5 July 2017).

### 2.2. Plasma Sample Collection

Peripheral blood was collected from unfasted participants at baseline and after 12 weeks of simvastatin treatment. Briefly, whole blood was collected into 10 mL EDTA-containing tubes and two-step centrifugation was performed (1600× *g* for 15 min, 5000× *g* for 10 min) to separate out the plasma. Plasma aliquots were stored at −80 °C until required for lipidomic analysis.

### 2.3. Plasma Lipidomic Analysis

Lipids were extracted from 10 µL of plasma mixed with internal standards, as listed in Huynh et al. (2019) [16], using a butanol/methanol extraction method [17]. Pooled human plasma from healthy individuals and NIST SRM 1950 human reference plasma were extracted and analysed together with the study plasma samples as quality controls.

Lipidomic analysis of the plasma lipid extracts was performed by liquid chromatography-mass spectrometry (LC/MS), using an Agilent 6490 QQQ mass spectrometer with an Agilent 1290 series HPLC system as previously described [16]. The concentration of lipid species was calculated by comparison with relevant internal standards and adjusted with response factors listed in Huynh et al. (2019) [16] [lipid concentration = (area of analyte/area of corresponding internal standard) × concentration of internal standard × response factor]. A total of 824 lipid species from 47 lipid classes were quantified. The concentration of the lipids in pmol/mL were normalised using the Probabilistic Quotient Normalisation method [5], and transformed to logarithm-2 for statistical analysis. 

Total cholesterol, HDL-C and triglycerides in plasma were analysed on a clinical panel accredited by the National Association of Testing Authorities using the COBAS 8000 modular analyser (Roche), with LDL-C estimated using the Friedewald equation [18]. 

### 2.4. Statistical Analysis

Statistical analyses were performed with R version 4.0.2. To determine if a patient has the circulating 3LS of poor prognosis at the time of blood collection, the lipidomic dataset was first aligned to the original cohort from which the 3LS was derived in Lin et al. (2017) [4] to adjust for batch differences, using the ComBat algorithm (R package sva, v3.34.0). The presence or absence of the 3LS for each plasma sample was calculated from the logistic regression model derived in Lin et al. (2017) [4] as follows: a patient was considered to have the 3LS when the probability of having the 3LS is greater or equal to 0.5 (*p* ≥ 0.5), where *p* is calculated from the logistic regression model as follows: *p* = e^y^/(1 + e^y^),


y = ln (*p*/1 − *p*)y = (3.1319 × ceramide(d18:1/24:1)) + 2.1724 × sphingomyelin(d18:2/16:0)) + (1.8593 × phosphatidylcholine(16:0/16:0)) − 91.217

Differences in lipid levels were assessed by paired sample *t*-tests (baseline versus post-simvastatin) or independent two-sample *t*-tests (samples with 3LS versus without 3LS)(R package rstatix, v0.7.0). *p*-values less than 0.05 were considered to be statistically significant.

## 3. Results

### 3.1. Cohort Characteristics

The recruitment rate was slower than anticipated due to the COVID-19 pandemic and the high prevalence of statin usage in men with prostate cancer. A total of 27 participants were prospectively recruited over 3 years between May 2018 to March 2021 (Figure 1B). An interim analysis was performed with these participants to determine whether there was any biological efficacy to continue with the study, and the findings are presented herein. Twenty-two of these patients provided paired plasma samples at baseline and post-simvastatin treatment (Figure 1B). Five men provided single time-point plasma samples (2 at baseline, 3 at post-simvastatin, Figure 1B). There were no side effects from the addition of simvastatin to standard therapy during treatment or the weeks following completion of simvastatin treatment.

The baseline clinical characteristics of the cohort are displayed in Table 1. Of note, most men were overweight or obese, with a median BMI of 28 (Q1 = 25, Q3 = 30). The median waist circumference was 106 cm (Q1 = 98 cm, Q3 = 111 cm), where a measurement of 94 cm or more indicates an increased risk of cardiovascular and metabolic diseases [19]. The number of men with diabetes, hyperlipidaemia or hypertension was 1 (4%), 2 (7%) and 9 (33%), respectively. Twenty men (74%) received taxane chemotherapy and 7 patients (26%) received ARSI.

The baseline total levels of cholesterol, LDL-C and triglycerides of all the participants, as measured by the clinical assay, were within the clinically recommended healthy range for individuals without cardiovascular risk factors. Mean baseline cholesterol was 4.87 mmol/L (reference range ≤ 5.5 mmol/L), mean baseline LDL-C was 2.87 mmol/L (reference range ≤ 3.0 mmol/L), and mean baseline triglycerides was 1.58 mmol/L (reference range ≤ 2.0 mmol/L). These results are in keeping with the men not being on statin therapy at the time of recruitment onto the clinical trial as they had no clinical dyslipidaemia detected by their primary care physician.

### 3.2. Effect of Simvastatin on the Poor Prognostic 3-Lipid Signature

The circulating poor prognostic 3LS derived previously [3,4,5] was used to represent the poor prognostic lipidomic profile of mCRPC. The 3LS was detected in baseline plasma samples of 11 of the 24 men with baseline samples (46%, Figure 1B), which is similar to the prevalence of ~40% that was observed in previous studies [3,5]. Additionally, consistent with previous studies on the 3LS [4,5], men with the 3LS at baseline had significantly higher plasma levels of sphingolipids than those without the 3LS at baseline (Figure 2A, Appendix A). These sphingolipids include monohexosylceramide, dihexosylceramide, trihexosylceramide and sphingomyelin (*p* < 0.01, Figure 2A, Appendix A). Although the total baseline levels of ceramide (Cer(d)) were not significantly higher, the individual baseline levels of several ceramide species were significantly elevated in men with the 3LS at baseline compared to those without the 3LS at baseline (*p* < 0.05; Figure 2A).

Of the 11 men with the 3LS at baseline, five of them (45%) lost the 3LS after simvastatin treatment, (i.e., the 3LS was not detected in their post-simvastatin plasma samples), whereas the other six retained the 3LS after simvastatin treatment (i.e., the 3LS was detected in their post-simvastatin plasma samples). Notably, the post-simvastatin plasma lipidomic profiles of men with the 3LS at baseline more closely resembled the baseline lipidomic profiles of men without the 3LS at baseline, as the levels of lipids (class total and individual species level) were not significantly different in such a comparison (Figure 2B, Appendix A). Acylcarnitines (AC) also appear to be reduced by simvastatin treatment as their levels were not significantly different in the comparison (Figure 2B, Appendix A). Acylcarnitines were associated with radiographic progression in mCRPC [20] and metastatic relapse in localised prostate cancer [5]. The levels of these lipids were higher in men with the 3LS compared to those without the 3LS (Figure 2A, Appendix A).

The response rate (loss of 3LS) of 45% to simvastatin was higher compared to our previous lipidomic study of matched baseline and end-of-treatment plasma samples from patients receiving standard treatment alone for mCRPC [20]. In that study, only three of 16 patients (19%) lost the 3LS at end of treatment [20]. All except two of the 16 patients were on first line treatment (taxane or ARSI). Notably, the end-of-treatment plasma samples were collected at radiological progression rather than specifically at a 12 week timepoint, where the time that the patients were on treatment ranged from 0.56–17.8 months (median 4.8). 

The men who lost the 3LS in the current study had higher BMI (1.2 fold, *p* = 0.02) and waist circumference (1.2 fold, *p* = 0.02) at baseline than the men who retained the 3LS. However, baseline cholesterol, triglyceride or LDL-C levels were not significantly different between the two groups of men (*p* > 0.377). The other baseline clinical characteristics or the type of standard treatment received with simvastatin were also not significantly different between the two groups of men (*p* > 0.318, Appendix A). 

Overall, these observations suggest that the addition of statin to standard therapy for mCRPC was able to “normalise” the poor prognostic lipid profile in men who had the 3LS prior to statin treatment.

### 3.3. Post-Simvastatin Lipidomic Changes in Men Who Lost the 3LS versus Those Who Retained the 3LS

All the participants reported full compliance in taking simvastatin while on the study. However, the levels of LDL-C, total cholesterol and triglycerides were not consistently altered after treatment in men who lost the 3LS after simvastatin treatment according to the clinical assay (*n* = 5, *p* > 0.05, Figure 3A). This occurred despite their probability of having the 3LS was reduced to less than 0.5 and the three lipids constituting the 3LS were mostly decreased after simvastatin treatment (Figure 3B, Appendix A). 

Closer examination of two of the men who lost the 3LS after simvastatin treatment showed that their LDL-C and total cholesterol level increased after simvastatin treatment. The three lipids constituting the 3LS were decreased after simvastatin treatment in these two men (Patient IDs 130410 and 130433, Appendix A) even though their LDL-C and total cholesterol levels increased, suggesting that simvastatin was sufficient to reduce their sphingolipid levels but not their cholesterol, and thus the resolution of the 3LS is independent of changes to cholesterol or triglycerides. These two men may be resistant to the cholesterol-lowering effect of simvastatin. In contrast, LDL-C and total cholesterol were decreased with simvastatin treatment in all the men who retained the 3LS after simvastatin treatment (*n* = 6, *p* = 0.003, Figure 3A, Appendix A), indicating that they had received sufficient simvastatin to decrease their cholesterol level but they were resistant to the sphingolipid-lowering effect of simvastatin. The underlying reason why these men retained the 3LS appears to be unrelated to the cholesterol-lowering effect of simvastatin.

Unlike the clinical assay, the LC/MS lipidomic assay showed that the levels of total cholesterol (sum of cholesteryl esters and free cholesterol) were reduced by simvastatin treatment for both men who lost or retained the 3LS after simvastatin treatment (Appendix A). However, the statistical significance of the change was less for men who lost the 3LS (−17.0%, *p* = 0.04) compared to men who retain the 3LS (−15.4%, *p* = 0.005), which may be related to the differences in the reduction in total cholesteryl esters and free cholesterol (Figure 4, Appendix A). Total triglyceride levels (TG(SIM), TG[NL]) were not significantly changed after treatment for either group of men (*p* > 0.05, Figure 4, Appendix A). Overall, changes in total cholesterol and triglyceride levels cannot be used as a surrogate to identify men who normalise the poor prognostic 3LS with treatment. 

In men who lost the 3LS after simvastatin treatment, there was significant reduction in their plasma levels of individual lipid species of sphingolipids including ceramides (−23 to −45%, *p* ≤ 0.046), hexosylceramides (−27 to −52%, *p* ≤ 0.049) and sphingomyelin (−28 to −44%, *p* ≤ 0.047) after treatment (Figure 4A). This was also seen in the total levels of some sphingolipid subclasses, with significant reductions in monohexosylceramide (−35.2%, *p* = 0.025), trihexosylceramide (−37.1%, *p* = 0.026), and sphingomyelin (−27.2%, *p* = 0.029) (Appendix A). The acylcarnitine levels were also reduced although the reduction was not statistically significant (−48.5, *p* = 0.064) (Appendix A).

In contrast, in men who retained the 3LS following simvastatin therapy (*n* = 6), there was no significant reduction in the total levels of sphingolipid subclasses after treatment (Appendix A), and the levels of only a few individual sphingolipid species were reduced (Figure 4B). 

In our previous study of men with mCRPC, we had identified 19 lipids that were associated with prognosis, of which 3 were used to derive the 3LS [4]. Of these 19 lipids, 18 were quantified in the plasma samples in this study. The levels of 9 prognostic lipids were significantly altered with simvastatin treatment in the direction of favourable prognosis—their levels were decreased after treatment if high baseline levels were previously associated with poor prognosis and vice versa (Figure 5, Appendix A). In contrast, only one of these 18 prognostic lipids were significantly altered after treatment in men who retained the 3LS after therapy but in the direction opposite to that of favourable prognosis (Figure 5, Appendix A).

### 3.4. Men Who Did Not Have the 3LS at Baseline

Thirteen men did not have the 3LS at baseline, and post-simvastatin plasma samples were only available from 11 of them (Figure 1). Four of these 11 men (36%) gained the 3LS after simvastatin treatment. The cholesterol and LDL-C levels of 10 of these 11 men were decreased after treatment (Appendix A). The plasma levels of sphingolipids were not significantly altered in these 11 men, except for the levels of hexosylceramides (HexCer, Hex2Cer, Hex3Cer) which tend to be increased in the men who gained the 3LS (not statistically significant, Appendix A). Overall, the reduction in cholesterol levels also do not seem to be related to the sphingolipid levels in men who did not have the 3LS at baseline. 

## 4. Discussion

Statins have been shown to reduce circulating levels of sphingolipids in non-cancer cohorts [7,8,10], but this is the first study to prospectively demonstrate that the addition of simvastatin therapy can reduce the circulating levels of sphingolipids in men with mCRPC commencing standard therapy (taxanes or ARSI), resulting in the ‘normalisation’ of their poor prognostic lipid profile. In this study, we found that 45% of the participants with the 3LS at baseline lost this poor prognostic biomarker after 12 weeks of simvastatin therapy. This rate of ‘normalisation’ of the lipid profile by the combination of simvastatin with standard therapy is an improvement from the 19% observed for patients receiving standard therapy alone [20]. Among the participants who did not have the 3LS at baseline, 36% gained the 3LS after simvastatin treatment. It is uncertain if their gain in 3LS is an indication of treatment resistance and related to their clinical outcome. Further follow-up and additional patient numbers are required to properly investigate this aspect. While numbers are small in this proof-of-concept study, these findings suggest that a poor prognostic lipid profile can be therapeutically targeted by metabolic therapies such as simvastatin.

There is epidemiological evidence linking statin use with improved outcomes across the clinical course of prostate cancer [11]. Meta-analyses and numerous population-based studies have shown that statin use is associated with a reduced risk of biochemical recurrence in localised disease [13,14], prolonged time to castration resistance in hormone-sensitive prostate cancer [21], and reduced prostate cancer-specific mortality [13,22,23,24,25]. Simvastatin is a lipophilic statin and it was selected for this study as it was more effective than hydrophilic statins (e.g., pravastatin) in reducing the metastatic process in an in vitro model of prostate cancer [26]. Furthermore, simvastatin at the dose of 40 mg daily was shown to lower plasma ceramides by about 25% [7]. However, we found that simvastatin did not reduce the plasma ceramides or other sphingolipids in men who retained the 3LS despite reductions in their cholesteryl esters. This indicates that their cholesterol metabolism, but not sphingolipid metabolism, is responsive to the mechanism of action of simvastatin. Changes in the other 18 lipids corresponding to the lost of 3LS was also not observed. 

Interestingly, men who lost the 3LS had higher BMI and waist circumference at baseline than those who retained the signature, although there were no differences in baseline triglycerides. It is not known if obesity is related to the efficacy of statin in reducing circulating sphingolipids. Overall, the reasons for the lack of changes in sphingolipid levels or 3LS to statin treatment in this group of “statin-resistant” men remains unknown, and may explain why there are some studies reporting that statin treatment did not improve clinical outcomes in mCRPC [27], i.e., perhaps not all men are responsive to the sphingolipid-lowering effect of statin treatment. Previous studies on the potential mechanisms underpinning the antineoplastic effect of statins in prostate cancer were focused on the ability of statins to promote apoptosis [28], and inhibit inflammation [29], angiogenesis [30], cell proliferation [31], migration, invasion [26], and hypoxic adaption of tumour cells [32]. In particular, the antineoplastic effect of statins on prostate cancer cells was thought to occur through: (1) cholesterol-mediated mechanisms (whereby cholesterol-rich lipid rafts in cell membrane are disrupted, affecting signal transduction); and (2) non-cholesterol-mediated mechanisms (affecting other signalling pathways such as Ras and Rho) [11]. Based on the findings of our study, we hypothesise that statins may also have an antineoplastic affect through its ability to significantly reduce the plasma levels of sphingolipids. In vitro and in vivo studies have demonstrated the role of sphingolipid metabolism in various cancers, including prostate cancer, through the ceramide-sphingosine-1-phosphate (S1P) signalling axis [33,34]. The metabolic conversion of ceramide to S1P induces tumour-promoting and pro-inflammatory properties [35]. Men with prostate cancer and elevated circulating ceramides may have enhanced ceramide metabolism and S1P signalling, which promotes tumour growth and therapeutic resistance [36]. Therefore, lowering the circulating levels of ceramides may suppress the ceramide-S1P signalling axis and improve patient outcomes.

The concept of repurposing drugs to identify new uses for already approved medications with known safety and efficacy profiles is not new. The increasing appreciation for the interplay between lipid metabolism and cancer biology generates novel metabolic therapeutic targets in oncology. This study has demonstrated the potential of repurposing the drug simvastatin, a readily available and relatively affordable medication used to treat hypercholesterolaemia in cardiovascular disease and metabolic syndrome, to pharmacologically target the poor prognostic plasma lipid profile in men with mCRPC. There are possible clinical challenges to the practical implementation of this therapeutic strategy. Firstly, the assay to identify men with the poor prognostic lipid profile for metabolic targeting—in this case the 3LS—needs to be accessible, accurate and rapid with an acceptable turnaround time for use in a clinical setting. 

Furthermore, the use of simvastatin as a novel drug may be an issue given the overlapping demographics of prostate cancer with hypercholesterolaemia and consequently the high prevalence of statin use [14]. Men with mCRPC, who are already on statin therapy but are found to have the 3LS, may be resistant to the effect of simvastatin on sphingolipids, and thus may benefit from another metabolic treatment targeting sphingolipids. For example, proprotein convertase subtilisin/kexin type 9 serine protease (PCSK9) inhibitors, a newer class of drugs used to treat hypercholesterolaemia, have also been shown to lower the levels of circulating ceramides and other sphingolipids [37], and can be safely combined with statins [38,39]. 

Limitations of our current study are the single arm and open-label setup, and the low number of patients due to poor recruitment caused by the COVID-19 pandemic and the high prevalence of baseline statin usage amongst men with mCRPC. Additionally, exercise and diet were not monitored, which may have an impact on the levels of circulating sphingolipids [40,41]. However, the amount of dietary changes and intensity of exercises that are required to significantly alter the circulating sphingolipids in humans is unclear. It is unlikely that the participants in our study had significant changes in their diet and exercise over the short intervention period. 

The effect of decreasing sphingolipids on prostate cancer growth and pathological changes was not determined from this study, as the study was not designed to do so. The primary outcome of the study was to determine if simvastatin can modulate the circulating sphingolipids. A prospective randomised clinical trial of the metabolic intervention with monitoring of clinical outcomes is more suited to address the question on whether modulation of circulating sphingolipid levels in patients with the poor prognostic profile can improve clinical outcomes. 

Further proof-of concept studies with other metabolic drugs such as PCSK9 inhibitors may determine if a better response rate than simvastatin can be achieved. Recruitment is likely to be easier given that PCSK9 inhibition can be safely combined with simvastatin, and PCSK9 inhibitors are not widely used in this population currently. Following the identification of a suitable metabolic intervention, randomised control studies would be performed to determine if modulation of the circulating lipidomic profile is associated with improved clinical outcomes. Patient selection will be enriched by the use of a biomarker such as the 3LS to select men most likely to benefit from metabolic targeting of a poor prognostic lipid profile.

## 5. Conclusions

Simvastatin in addition to standard treatment for mCRPC can modulate the circulating lipidomic profile by reducing plasma levels of sphingolipids, and eliminating the presence of the 3LS at twice the expected response rate. The next step is to determine whether modulation of the circulating lipidomic profile is associated with improved clinical outcomes. 

## Figures and Tables

**Figure 1 cancers-14-04792-f001:**
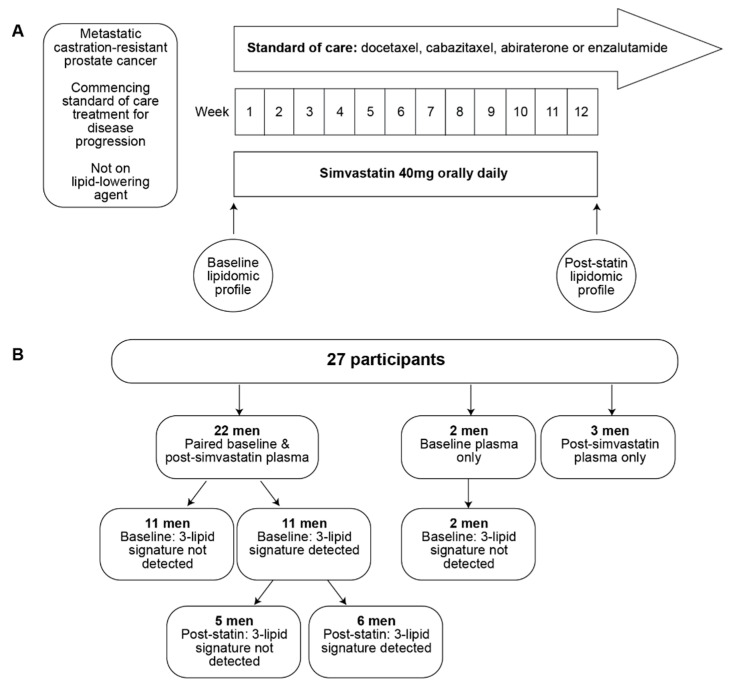
Study schema and participants of the clinical trial: (**A**) Study schema; (**B**) number of participants and their 3-lipid signature status.

**Figure 2 cancers-14-04792-f002:**
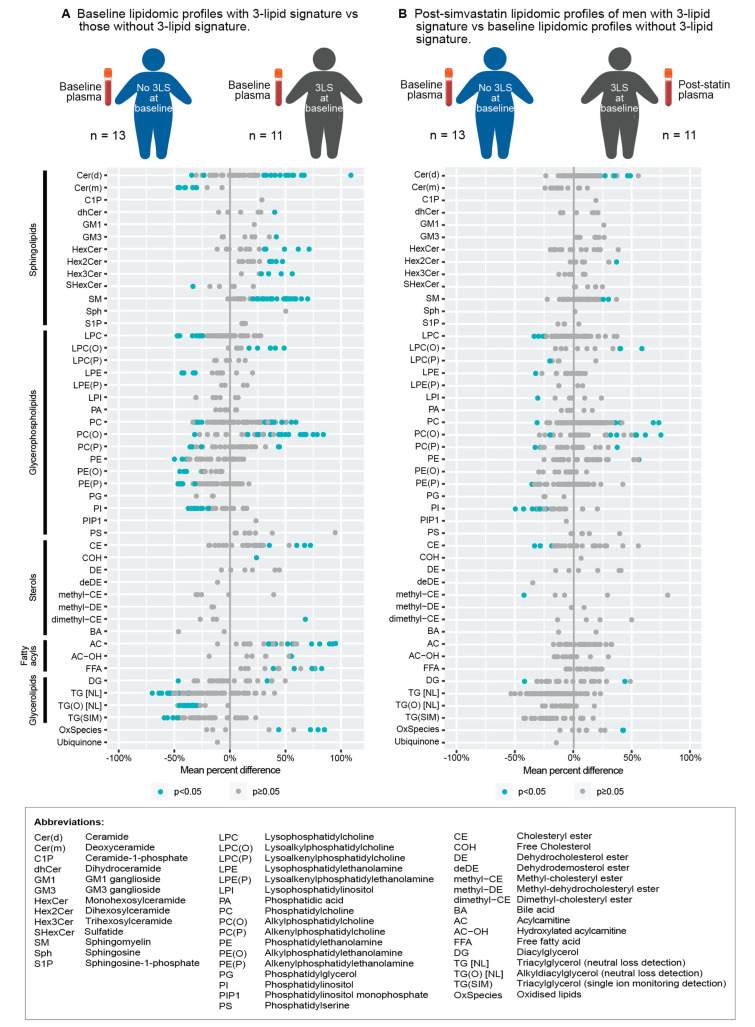
Mean percent difference of plasma levels of individual lipid species, grouped by lipid class, in two comparisons: (**A**) baseline lipidomic profiles with 3-lipid signature vs. those without 3-lipid signature; and (**B**) post-simvastatin lipidomic profiles of men with 3-lipid signature vs. baseline lipidomic profiles without 3-lipid signature. Each datapoint represents a lipid species within the lipid class. Blue datapoints indicate statistically significant differences with *p*-value < 0.05 by *t*-tests.

**Figure 3 cancers-14-04792-f003:**
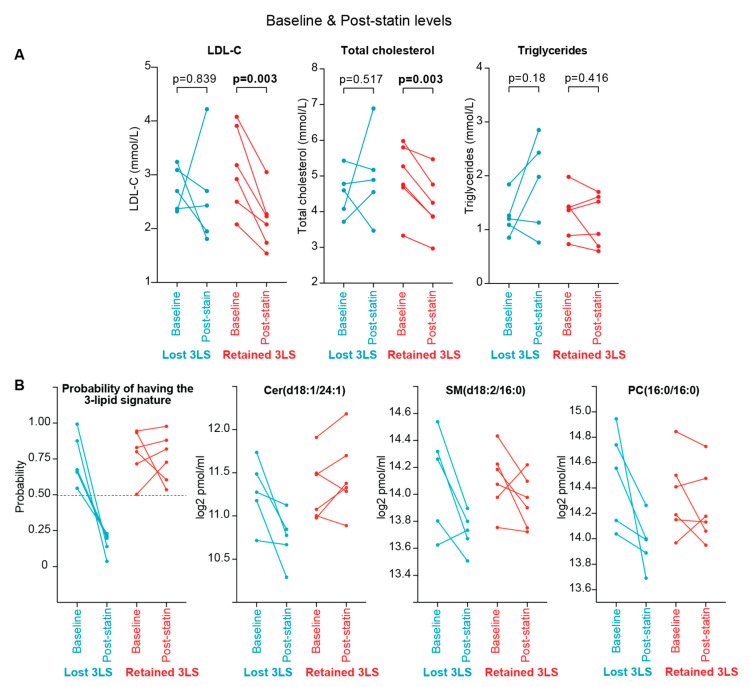
Baseline and post-simvastatin status of clinical lipids and the 3-lipid signature in men who lose the 3-lipid signature and those who keep the 3-lipid signature: (**A**) plasma levels of LDL-C, total cholesterol, and trigylceride levels, as measured on a clinical assay (paired t-test); (**B**) probability of having the 3-lipid signature (calculated by the logistic regression model), and plasma levels of the three lipids of the 3-lipid signature.

**Figure 4 cancers-14-04792-f004:**
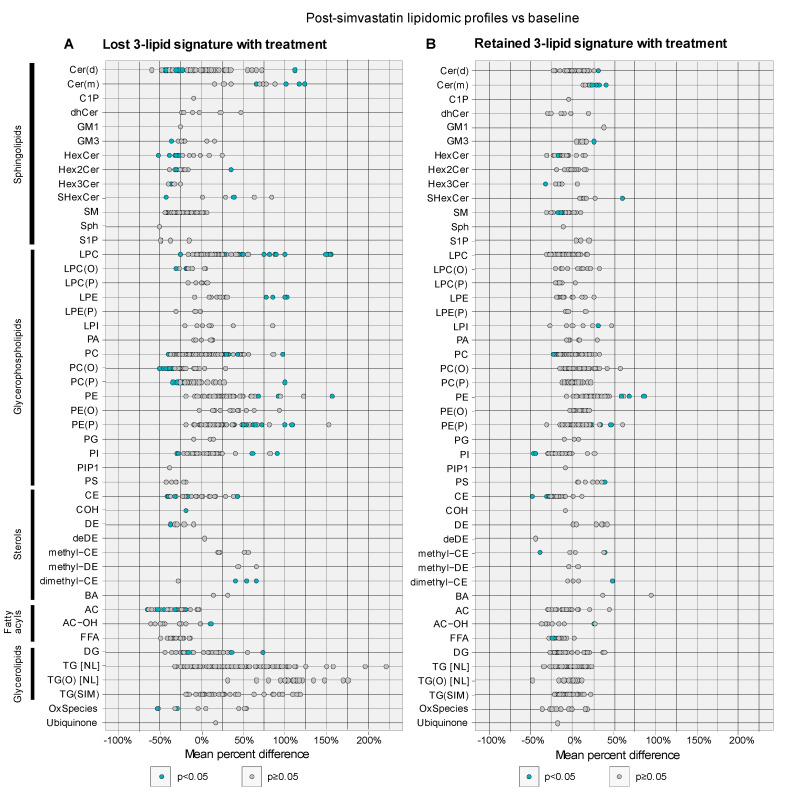
Mean percent difference of plasma levels of individual lipid species, grouped by lipid class, in post-simvastatin profiles vs. baseline from (**A**) men who lose the 3-lipid signature with treatment; and (**B**) men who keep the 3-lipid signature with treatment. Each datapoint represents a lipid species within the lipid class. Blue datapoints indicate statistically significant differences with *p*-value < 0.05 by paired *t*-tests.

**Figure 5 cancers-14-04792-f005:**
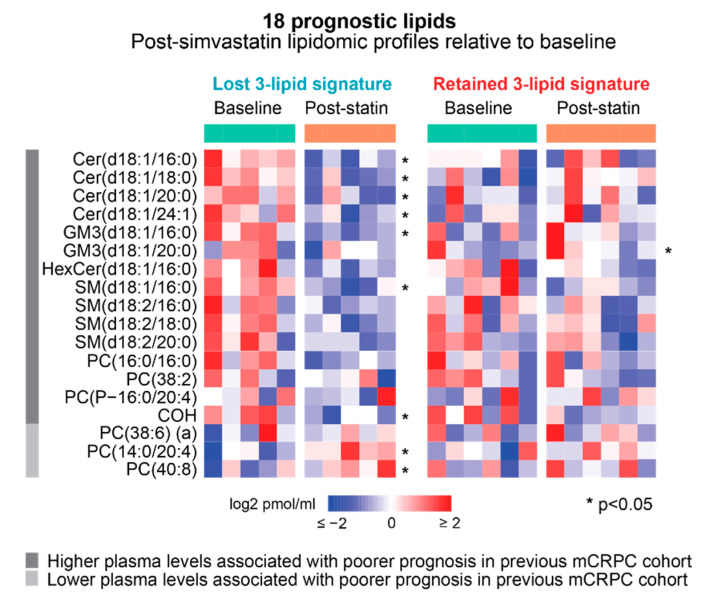
Heatmaps of change in the levels of 18 prognostic lipids in post-simvastatin lipidomic profiles relative to baseline, in men who lose the 3-lipid signature and those who keep the 3-lipid signature. * Significant differences, *p*-value < 0.05 by paired t-test. Prognostic association in previous mCRPC cohort refers to Lin et al. (2017) [4].

**Table 1 cancers-14-04792-t001:** Baseline clinical characteristics of the cohort and standard treatment received with simvastatin.

Characteristic	Median (1st Quartile, 3rd Quartile) or Number (%)
Age (years)	74 (72, 79)
ECOG performance status:	
0–1	26 (96%)
≥2	1 (4%)
Gleason grade at diagnosis:	
<8	5 (18.5%)
≥8	15 (55.5%)
Unknown	7 (26%)
Site of metastasis:	
Lymph node	17 (63%)
Bone	23 (85%)
Visceral	2 (7%)
Prostate-specific antigen (ng/mL)	42 (17, 77)
Alkaline phosphatase (U/L)	87 (75, 171)
Lactate dehydrogenase (U/L)	256 (206, 287)
Haemoglobin (g/L)	127 (121, 138)
Metabolic risk factors:	
Diabetes	1 (4%)
Hyperlipidaemia	2 (7%)
Hypertension	9 (33%)
BMI	28 (25, 30)
Waist circumference (cm)	106 (98, 111)
Treatment with simvastatin:	
Docetaxel	17 (63%)
Cabazitaxel	3 (11%)
Enzalutamide	5 (19%)
Abiraterone	2 (7%)

## Data Availability

Data used in the analysis of this manuscript may be shared upon request. Please contact the corresponding author.

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
