# Peer review of "Modulation of Plasma Lipidomic Profiles in Metastatic Castration-Resistant Prostate Cancer by Simvastatin"

_cancers, 2022, doi:10.3390/cancers14194792_

Round 1

Reviewer 1 Report

The study by Lisa Horvath and team details the impact of simvastatin intake on plasma lipidomic profile of men with metastatic castration-resistant prostate cancer (mCRPC). The study primarily focuses on the changes in the validated 3-lipid signature (baseline) and after 12-week simvastatin intervention. The authors demonstrate that simvastatin was able to reduce plasma sphingolipids in the mCRPC patients (independent of changes in plasma cholesterol, triglycerides etc) and infer that the decrease in sphingolipids could be beneficial against prostate cancer growth. The study is well-designed, and the inference drawn is supported by the data outcomes.

Minor Comment: Though the authors detail the impact of simvastatin on circulating sphingolipids, the direct correlation between decrease in sphingolipids by simvastatin intake and effect against prostate cancer growth is hypothetical and not investigated in the study. The reviewer considers this as a study limitation; to address this limitation [given that no prostate cancer biomarkers, biopsy or pathological changes (tissue level)] were included in the study to indicate direct correlation, the authors may want to discuss these limitations in the discussion section and share aspects of future work that can address this question.

Reviewer 2 Report

Mak et al. reported the results of the single arm proof-of-concept trial where simvastatin was given as an adjuvant therapy for 12 weeks with standard treatment for men with metastatic castration resistant prostate cancer. This is an interesting and novel study, however, I have several comments about the set-up and results.

1.     In the introduction, it may be a bit clearer to readers who are not that familiar with different classes of lipids if authors add some information on what kind of lipids are included to the 3-lipid signature. For instance, only two of them belongs to sphingolipids.

1.     Baseline characteristics for men with and without 3LS should be listed too (maybe table to supplemental material). Were there any differences between the men who lost their 3LS and retain 3LS in background characteristics? Of course, with n-value of this low, it is impossible to draw any conclusion about this, however, it would be interesting to see whether responders were on different adjuvant therapy or had lower or higher BMI etc. compared to non-responders.

2.     Did you follow the possible side-effects caused by statin treatment? Those should be reported. Is this trial registered to anywhere (e.g. clinicaltrials.gov)?

3.     Just by looking at Figure 2, it seems that statin treatment decreases upregulation of lipid classes of acylcarnitines, hydroxylated acylcarnitines, and free fatty acids. Did you looked the role of these classes or some individual lipid species in the patients who responded to statin treatment? Based on previously published studies, have these lipid classes any prognostic value or have those linked to cancer development or progression?

4.     Did any of the no 3LS subjects at the baseline had 3LS after simvastatin treatment? This should be mentioned it the manuscript. Furthermore, in the result section it should be mentioned what happened cholesterol levels in this group after statin treatment?

5.     In general, power of this study is too low for any conclusion. Did authors conduct any power calculation beforehand for this single arm proof-of-concept trial? If yes, this should be mentioned in the manuscript. Also, authors should soften the conclusions about the effects of statin treatment on 3LS because only 45% of the subjects with it responded to the treatment, especially if it is estimated that within subjects with 3LS around 10-20% will lost this signature anyway after standard mCRPC treatment.

6.     In lines 208-223 authors speculate the sufficient simvastatin dosing. Does this mean that subjects did not use simvastatin as instructed or those two patients were resistant to simvastatin treatment and should use higher dose? Did authors calculate compliance of simvastatin use?

7.     In the discussion, dosing of simvastatin should be mentioned. In this trial 40 mg daily dosing was use, however, even 80 mg daily dosing is possible with statins, thus, it may also explain the differences in responses to statin treatment between the subjects.

8.     How much diet and other life-style related factors have impact on lipidome in serum, especially for ceramides and sphingolipids? Without registering died it is impossible to say anything about this topic in this trial, however, it might be worth of mentioning that this aspect might highly impact on serum lipidome and lipid-lowering drug treatment responses.

9.     In the discussion, limitation of the current trial should be mentioned, e.g. low number of patients, single arm and open label setup, no monitoring of the diet etc.

Minor comment:

Some symbols in the figures (at least in my pdf version) are nor visible. These should be checked and correct.

Inclusion and exclusion criteria should be listed in the materials and methods

Round 2

Reviewer 2 Report

I am satisfied for the authors reply to my comments, thank you!